# Examining Clinical Practice Guidelines for Exercise and Physical Activity as Part of Rehabilitation for People with Stroke: A Systematic Review

**DOI:** 10.3390/ijerph19031707

**Published:** 2022-02-02

**Authors:** Gavin Church, Ali Ali, Christine Leslie Smith, Dave Broom, Karen Sage

**Affiliations:** 1Community Stroke Service, Sheffield Teaching Hospitals, NHS Foundation Trust, SPARC, Department of Health and Wellbeing, Sheffield Hallam University, Collegiate Cres, Broomhall, Norfolk Park Road, S2 3QE, Sheffield S10 2BP, UK; 2Stroke Consultant and Stroke Research Lead. L Floor, Royal Hallamshire Hospital, Sheffield Teaching Hospital NHS Foundation Trust, Broomhall, Glossop Road, Sheffield S10 2JF, UK; ali.ali9@nhs.net; 3Department Allied Health Professions, Sheffield Hallam University, Collegiate Cres, Broomhall, Sheffield S10 2BP, UK; chris.smith@shu.ac.uk; 4Research Centre for Sport, Exercise and Life Sciences, Coventry University, Priory Street, Coventry CV1 5FB, UK; ad5173@coventry.ac.uk; 5Applied Clinical Research, Department of Nursing, Manchester Metropolitan University, Brooks Building, Manchester M15 6GX, UK; K.Sage@mmu.ac.uk

**Keywords:** stroke, exercise, guidelines, systematic

## Abstract

Background: Stroke is the leading cause of chronic physical disability in Western industrialised nations. Despite clear guidelines for exercise in individuals with many non-communicable diseases, the guidance for people with stroke (PwS) who frequently present with multiple comorbidities is less clear. A systematic review of exercise guidelines was undertaken to synthesise themes and patterns. Methods: The review was completed according to the PRISMA statement. Guideline-specific databases were searched for worldwide clinical practice guidelines (CPGs). All included papers underwent quality assessment using the AGREE II protocol. Content synthesis and analysis of the guidelines was undertaken using CERT. Results: Searching identified 2184 papers. After duplicate removal and screening by title and abstract, 22 CPGs remained for review. Seven guidelines identified three key roles for exercise interventions: (1) promoting a healthy lifestyle, (2) prevention of further strokes and (3) rehabilitation. Of concern, many CPGs fail to recommend appropriate safety measures and standards, pre-, during and post-exercise or tailor for specific needs. Conclusions: Global guidelines for exercise in PwS lack in-depth and technical information on the exercise delivery methods, application and dosage required to progress exercise interventions for PwS.

## 1. Background

Stroke is the leading cause of chronic physical disability in Western industrialised nations [1]. In the European Union (EU), stroke is the second most common cause of death and a leading cause of adult disability, with cardiovascular disease (CVD) being the largest underlying cause [2]. It affects 1.1 million inhabitants of Europe every year and causes 440,000 deaths, and by 2047 Wafa et al. (2017) estimates an additional 40,000 incident strokes and 2.58 million prevalent cases. Conversely, 80,000 fewer deaths and 2.31 million fewer disability-adjusted life years lost are projected [2]. Published Clinical Practice Guidelines (CPGs) provide clinicians, health care practitioners and patients with a summary of information regarding clinical conditions. Bradshaw (2017) [3] identified how CPGs are systematically developed statements to assist practitioner decisions about appropriate health care and how they can be used to reduce inappropriate variations in practice and to promote the delivery of high quality, evidence-based health care. 

The role of exercise and physical activity can play an important role in preventing and managing several long-term health conditions such as coronary heart disease, type 2 diabetes, stroke, mental health problems, musculoskeletal conditions and some cancers [4]. It also has a positive effect on wellbeing and mood, providing a sense of achievement or relaxation and release from daily stress [4].

There are clear and consistent guidelines for exercise in the non-clinical adult population (18–65 years and those over 65 years) in Europe and worldwide (Public Health England 2016 [4]/WHO (2020) [5]/European Commission (2018) [6]). These advise on the amount and intensity of physical activity and exercise as part of daily living that will benefit health. There is a lack of clarity in the guidelines about the type of exercises that should be used as part of balance and strength training, how to achieve sufficient benefit from exercise training, what might be a sufficient intensity and the threshold to deliver changes in health physiology. For clinical and non-clinical populations, this is particularly important to understand if an aim of exercise is to aid in the neurological recovery [7] and structural soft tissue adaptations for strength and cardiovascular fitness [8]. 

There is strong public health evidence that exercise and physical activity in adults can reduce the primary risk of CVD and stroke by 20–35% [4]. Exercise post-stroke is now established as beneficial for secondary prevention, physical recovery, neuroplasticity and psychosocial adjustment [9]. Exercise interventions aimed specifically at people with stroke (PwS) and CVD aim to (a) reduce further stroke and CVD risk, (b) promote a healthy lifestyle and (c) yield rehabilitation benefits through improvement to fitness and strength. This links improvements in physical activity with increased engagement and participation in activities of daily life [10]. When the intensity of exercise is used as a means of ensuring an exercise stimulus or threshold is achieved, this is reflected in better daily life activity outcomes [10,11]. Despite this, established group intervention schemes such as cardiac rehabilitation may lack sufficient intensity to create an exercise stimulus for its participants [12]. 

Despite clear guidelines for exercise with non-clinical groups and individuals with long-term conditions such as diabetes, lung conditions and heart failure, the guidance for PwS who frequently present with multiple comorbidities is less clear [13]. In particular, the components of exercise that address sensorimotor loss and global deconditioning, identified by the Stroke Association Exercise and Activity Guidelines as required elements in the exercise programmes, are missing [14]. UK guidelines from the National Institute for Health and Care Excellence (NICE) [10] and Royal College of Physicians (RCP) [11] provide an overview for health care practitioners of the need for exercise in PwS but lack depth and precision. Other guidelines and proposed frameworks, from the American Heart Association/American Stroke Association (2004) [14] and best practice guidelines for exercise in stroke from the University of Glasgow and Caledonian (2010) [15] provide more depth, but the quality of primary studies used has not been reviewed as rigorously. Most of these guidelines focus on exercise interventions aimed at PwS who can walk because exercise practice in this subgroup of stroke patients may be protective against future decline and/or further comorbidities [16]. Adopting more depth and guidance on exercise and physical activity not only supports clinicians to deliver intervention to reduce the risk of further stroke, but also helps reduce the frequency of falls and the demand placed on the National Health Service (NHS) and social care resources [17].

It is unclear why uptake and delivery in clinical practice such as mainstream NHS rehabilitation settings and exercise on referral schemes is so varied; Condon et al. in (2017) [18] identified the lack of exercise delivery experience in the health care practitioners who provide exercise interventions to a clinical group that they perceive to be vulnerable.

A systematic review was undertaken, the aim of which was to identify and analyse CPGs for PwS, and to understand how they are developed [19]. In particular, we sought to examine what is consistent across guidelines, how guidelines compare in terms of depth and precision of guidance provided, whether there are areas that are clearly understood and easily implemented and, finally, whether they have considered training needs and resources in relation to exercise with PwS.

## 2. Methods

### 2.1. Registration, Selection Criteria

This review was registered with Prospero and accepted in September 2020 (REF:172057). A systematic review was chosen to explore what exercise guidelines for PwS are available from around the world, the information they provide to professionals, and any barriers or limitations that impact on the delivery and uptake of the information presented [19]. These guidelines in PwS support the investigation of new professional roles and help inform the prioritisation of services where evidence on exercise effectiveness is equivocal [20,21]. The summary style information collected from CPGs meant a meta-analysis was not appropriate as part of this review, as quantitative information was not always available, even when requested by corresponding authors. 

This review included global CPGs only that were specific to PwS, or exercise-related guidelines including the stroke population, and were only included if they were aimed at adults over the age of 18 years.

CPGs that did not provide guidance in the prescription of exercise or were about therapy only, and advice only, were excluded, as were those devoted solely to healthy lifestyle or adjuncts to exercise such as health eating or physical activity advice based on government recommendations. The full inclusion criteria are provided in Table 1.

### 2.2. Databases

The following databases were searched:National Institute of Health and Care Excellence (NICE)Turning Research Into Practice (TRIP)Scottish Intercollegiate Guideline Network (SIGN)International Guidelines LibraryBritish Medical Journal database (BMJ)

Two medical databases, Medline (EBSCO), and CINAHL Complete (EBSCO), were searched after the searches of the above guidelines database were complete to ensure no further guidelines had been missed. The International Guideline Clearing House Database was not searched because of its heritage/ghost site status (it has not been updated since 2017). All of the above searches were undertaken in September 2021 and exported into an Excel spreadsheet where duplicates were removed. Guidelines that had been superseded with more recent versions were also removed. 

### 2.3. BMJ and International Guidelines Search Method

Once these searches were completed, the BMJ database and International Guidelines Library was searched.

When the BMJ database is searched, it identifies the most relevant journals in order and based on the relevance to the keywords used in the search. In order to maximise the chances of finding anything not previously found, a hand searching strategy was used where the first author searched the first 100 suggestions for suitable papers and then repeated the process for the next 100, stopping once no suitable papers appeared in a set of 100.

The International Guidelines Library database does not enable searches to be exported, so this was searched manually and guidelines were exported by hand.

### 2.4. Search Strategy

A building-block approach from Booth 2008 [22] identified search terms for each concept, as advised by a university information scientist. The concepts were Exercise (Concept A), Stroke (Concept B), and Guidelines (Concept C). Using this, the search strategy comprised: MeSH terms to describe stroke (Stroke+ or Cerebral Haemorrhage+ Stroke* or CVA or cerebrovascula* acciden*), MeSH terms to describe exercise (Exercise+ or Activity+), and MeSH terms to describe guidelines (Guideline+ or Framework+). 

Key words were not used as the specific guideline databases used in the search were not able to handle a breakdown of keywords like in standard research databases such as PubMed or Ovid. The Boolean operators AND and OR were used, alongside phrase, proximity and truncation operators dependent on the database used. The search syntaxes were adapted accordingly for each information source and controlled vocabulary terms were used where available. 

### 2.5. Screening and Selection Process

Screening was undertaken using the inclusion and exclusion criteria set out in Table 1. The title and abstract, forward, introduction or preface were screened by the first author (GC) and ten percent of the included and excluded papers were checked by a second reviewer (AA). Any disagreements were mediated by a third author (KS).

Full text articles identified from the screening process were reviewed by the first author (GC), with ten percent reviewed by the second author (AA) and any disagreements were reviewed by the third author (KS).

#### 2.5.1. Quality Assessment

All included papers underwent quality assessment using the Appraisal of Guidelines for Research and Evaluation version 2 (AGREE II) protocol designed for use in the review of guidelines. The AGREE II protocol and checklist has been identified as a credible tool used in evaluating the bias in CPG [23]. If data were missing that were necessary to complete the quality assessment, the corresponding author for the guideline was contacted by email.

Quality assessments were undertaken by the first author (GC) and checked by a second author (AA). The tool involves a 1–7 rating scale representing strongly disagree to strongly agree, respectively. Papers were not excluded on the basis of the quality assessment.

There was a single disagreement when assessing the quality of the RCP National Clinical Guidelines for Stroke 5th Edition. Section 5 scores the views and preferences of the target population (patients, public, etc.). The first reviewer (GC) initially scored it as 7, but after discussion with AA this was reduced to a 6. This was due to the RCP identifying the role of external reviewing using laymen and working groups, but there was no mention of the working group involved patients. 

#### 2.5.2. Data Extraction Method

Extracted data included authors, year of publication, year due for review and the type of guideline. The outcomes captured from papers included the exercise types promoted such as walking, circuit class or hydrotherapy. The delivery of the exercise was extracted and included sets, repetitions, frequency, duration and the instructor’s role, qualification, and responsibility. The setting used for delivery was extracted, such as hospital, community leisure centres or private gyms. Safety precautions such as pre-assessment screening or checks to cardiovascular parameters before, during and after exercise were captured to assess the safety precautions used. 

Content synthesis and analysis of the guidelines was undertaken using the CERT. This tool enables transparency, improves trial interpretation and replication, and facilitates implementation of effective exercise interventions into practice [24,25]. Data from the identified articles were completed using the CERT and extracted to a bespoke electronic repository. CERT uses a scoring system where 0 = not included and 1 = included. The CERT scoring method was piloted by 2 reviewers (G.C. and K.S.) who, after piloting three articles, added a 0.5 score to reflect information that was included but that lacked depth.

#### 2.5.3. Analysis and Synthesis

Narrative synthesis, with supporting tabular synthesis, was used to interpret the findings from this review. This included the quality of the research used within the guidelines using the AGREE II protocol, including reasons for variations in quality. The homogeneity and heterogeneity of the reviewed guidelines was extracted to allow comparisons of consistency and variation of guidelines. Content analysis was collected using the CERT and exposed the clarity of exercise guidance for PwS and practitioners. 

## 3. Results

The preliminary search from all databases identified 2184 articles. After duplicates were removed and five additional articles were added from the International Guidelines Library, 1546 articles remained for screening by title. Of these, 1501 were eliminated by title, leaving 45 for review by abstract. Of these, 23 were excluded due to: two not being CPGs, five were non-stroke guidelines, 15 were not specific to stroke and one was a guideline providing advice only. This left 22 CPGs for review.

At each stage of the exclusion process, ten percent of article examples from the exclusion and inclusion process were sent to a second reviewer for checking. A full breakdown of the PRISMA flow diagram is given in Figure 1.

Table 2 shows all the CPGs and their country of development. It also includes the objective of the guidelines in relation to exercise prescription as a means of rehabilitation, health lifestyle guidance, further stroke prevention or a combination of these. Of the 22 CPGs included and reviewed: seven were from America [14,26,27,28,29,30,31], five were from England [10,11,32,33,34], three from Canada [35,36,37], and one each from: Scotland [15], the European collaborative (multiple countries in Europe collaborating) [38], Spain [39], Wales [40], Malaysia [41], The Netherlands [42], and Australia [43].

Seven identified the role of exercise intervention in health, prevention and rehabilitation, ten in the role of health and prevention only, three for prevention only, and two for health only. Sixteen of the guidelines were specific to PwS and three were exercise/physical activity specific to adults, but included references relevant to PwS and the clinical management of stroke. 

The Agree II protocol is sectioned into six domains, scope and purpose, stakeholder involvement, rigour of development, clarity of presentation, applicability and editorial independence.

All of the guidelines scored positively for the overall scope and purpose of the guideline, suggesting a low risk of bias in their development. Most of the guidelines demonstrated a clear role for stakeholders and the processes they were involved in. The involvement and role of the specific clinical group was not clear in half of the guidelines. The guidelines that did identify the involvement of the clinical groups were not clear about how this group was used in developing the guideline [14,15,27,29,35,38,40,42,43].

National guidelines for stroke management appeared to demonstrate strong rigour in their development [10,27,28,29,32,33,34,38,41,42,43]. Guidelines more specific to exercise interventions appeared to demonstrate less rigour or did not identify the systematic processes used to demonstrate the rigour of the development [14,32,36].

All of the guidelines demonstrated a positive clarity of presentation and ability to identify key findings and recommendations. Most failed to positively facilitate the application of the guidelines in clinical practice. CPGs specific to exercise interventions appeared to consider the barriers, facilitators, and strategies to apply guidelines based on current resources.

A strong editorial independence and roles were evident in most national guidelines. One guideline lacked clarity about the roles the internal and external reviewers played and failed to identify any declaration of interests [40]. Four of the guidelines failed to identify these points at all, therefore challenging the risk of bias in these CPGs [14,35,38,41]. The Netherlands guidelines [42] were the only guidelines to mention how they were developed alongside the Agree II protocol to ensure optimal quality was achieved. 

A full breakdown of the quality assessment is given in Table 3. 

The CERT contained six domains consisting of materials/equipment used, provider information and training, location, dosage, general application and tailoring needs, and how well they are planned and executed. 

Most of the guidelines were consistent in identifying the types of exercise in some detail but only eight demonstrated multiple options for exercise intervention and the equipment required [14,15,34,35,36,40,42,43]. Three of the guidelines discussed the appropriate qualification, skills and experiences needed for running an exercise intervention in PwS. One guideline conveyed more depth with options for qualifications, competencies and future training options to maintain these skills. It also offered options for non-exercise specific courses that may support motivation and behaviour uptake in exercise interventions [15]. The Malaysian guidelines [41] only discussed exercise in relation to specific joint exercises or mobility therapy performed by a therapist on a person and had no mention of exercise and its role in health, prevention or rehabilitation. 

Location was discussed in brief by eight of the papers [14,15,34,35,36,39,42,43]. There was no justification into the benefits, barriers, strengths or weaknesses of these options. Most guidelines discussed in some depth the use of dosage and methods of altering the dosage. Some guidelines [14,15] discussed in depth the Frequency, Intensity, Time and Type (FITT) principles for modifying exercise. It was unclear in most of the guidelines if there was a methodology to the FITT principle for clinicians to follow or how components such as intensity were measured or prescribed appropriately. 

Apart from the Malaysian guidelines, all the guidelines discussed in some depth the generalisable qualities of exercise in PwS as a means of health and prevention. Only half of the guidelines [14,15,27,34,35,36,38,39,40,42,43] briefly identified the need to modify or tailor exercises, specific to the individual needs. Three of the more specific guidelines for exercise in its role as a rehabilitation intervention discussed the need to tailor exercise based on an individual’s level of skill, mobility, baseline level of fitness and appropriate outcome measures [15,36,42]. 

Only five of the guidelines [15,34,36,42,43] identified the role of reflecting on how well exercise intervention was delivered and received by the target group as part of intervention fidelity. This domain of CERT generally lacked depth in all the guidelines. Table 4 provides a full breakdown of the data extracted from CERT.

Four of the guidelines [15,35,42,43] acknowledged the need for pre-exercise safety checks using heart rate and blood pressure measurements. Of these, only one guideline [35] discussed the role of pre-, during and post-safety checks and the use of pre-assessment fitness markers as means of safety and outcome measurement. A full breakdown is provided in Table 4. 

## 4. Discussion

To the authors’ knowledge, this is the first systematic review of CPGs relating to exercise and physical activity interventions in PwS. Guidelines are intended to provide those who need them (e.g., health practitioners, clinical groups, exercise practitioners) with a summarised package of information for the management and application of certain interventions. When the complexities within clinical groups such as individuals with cardiac failure, diabetes or stroke are considered, a single guideline does not provide a one-size-fits-all guide in managing those complex, different conditions. National guidelines have to cover aspects of condition management, prioritise interventions based on the evidence and the impact they can have on the clinical groups, and then present the information at a relatively superficial summary level rather than at a depth that provides enough detail for clinical practice. The AGREE II protocol identifies that guidelines should also provide information on the level of stakeholder involvement and application of these guidelines and the necessary resources to achieve this [23].

A range of clinical guidelines for exercise intervention in PwS was identified and reviewed. Apart from the Malaysian stroke guidelines [41], all of the other guidelines, originate from the Global North (e.g., Europe, USA, Australia and UK) which restricts the guidance for PwS to wealthier countries and does not address how stroke and exercise interventions may be managed in the Global South where health and wellbeing services are differently constructed. The lack of diversity of guidelines is particularly important as two-thirds of strokes occur in the Global South [44]. Poungarvin et al. (1998) [45] identified how the Asia-Pacific Consensus Forum on Stroke Management predicted that “in the next 30 years the burden of stroke will grow most in developing countries rather than in developed countries”.

The guidelines reviewed identify specific objectives for exercise interventions in PwS. These include guidance for a healthy lifestyle, exercise as a means of further stroke prevention, and exercise as a tool for further rehabilitation. Many of these guidelines used a combination of these objectives. Due to the scoring system, overall guidelines assessment and domain set-up in the AGREE II tool, analysing bias is not as clear is it is for primary research studies. Domains such as stakeholder involvement, rigour of development and editorial independence may have stronger links to bias compared to clarity of presentation. 

This review found that the national guidelines appear to be rigorously developed and have a low risk of bias (e.g., the RCP guidelines [10] and the AHA/ASA guidelines [26,27,29,42,43]. This was ascertained from scoring maximum marks on the rigour of development section of the AGREE II tool. Guidelines more specific to exercise intervention such as the Best Practice Guidance for the Development of Exercise after Stroke Services in Community Settings [15] appear to be less rigorously developed and had less depth regarding the systematic processes used in their development, thus increasing the potential risk of bias. Despite this, these guidelines provided more depth about the role and use of exercise in PwS and prioritised clear reporting in the development of the exercise intervention, the delivery and the evaluation processes. Although some variation in specific exercise content was acknowledged, all of the guidelines provided a clear scope and objectives for their use including appropriate health questions in PwS. The roles of the stakeholders were clearly defined in the majority of the guidelines, as was the involvement of Public and Patient Involvement (PPI). However, despite this being mentioned in some of the guidelines, the actual role and level of involvement was not discussed in any depth. These statements are critical to allow the target audience (e.g., practitioners and patients) to critically review the processes for evidence collection, see clear links to the recommendations and easily identify key recommendations. 

Despite CPG’s providing clear recommendations, there was a lack of detail to support the facilitation and application of the guidance given. This may be due to geographical variation in specific exercise interventions and local barriers which require specific strategies and facilitation plans to be made. These strategies should be based on resources, the location of the intervention, staffing availability and demand from the clinical group. The lack of clarity regarding PPI involvement in guideline development can be essential when reflecting on the engagement with guidelines into PwS. This may also be a contributing factor that causes poor uptake and engagement in aspects of guidelines’ use, particularly exercise intervention for PwS, which has been identified as a key issue in current literature [46,47]. 

Key findings from all of the guidelines reviewed are consistent with the national advice for exercise in non-clinical groups aged 18–65 years and those over 65 years, of 30 min of moderate intensity exercise at least five times per week [4]. Most guidelines provide some detail about the methods of exercising and the equipment commonly used but there appears to be a lack of depth into the justification and selection process of these for PwS. The generalisable needs of the clinical group and the need to tailor exercise in interventions is acknowledged in some of the more exercise specific guidelines [15,36]. 

Only three of the guidelines [15,34,36] discussed the appropriate qualifications, skills and experiences needed for implementing an exercise intervention in PwS. The Best Practice Guidance for the Development of Exercise after Stroke Services in Community Settings [21] discussed the options for qualifications and competencies. Within clinical practice it is essential for staff to receive the appropriate training and the opportunity to maintain these skills, also considering the psychosocial and coaching aspects of intervention delivery that may support motivation and behaviour uptake in exercise interventions. Simpson et al. (2013) [48] explored the willingness and confidence of allied health care professionals to prescribe exercise intervention appropriately in clinical groups. It was evident that an appropriate balance of knowledge, experience and self-competence is needed to successfully prescribe exercise in clinical groups. This was due to a concern that clinicians may cause harm to the individual, which in turn was concluded to be one of the primary barriers and facilitators supporting exercise prescription in clinical groups.

The location of exercise intervention highlights the benefits and drawbacks to particular locations based on practicality and the perception of the clinical group explored [46,47]. It has been identified that exercise should take place in a de-medicalised environment; however, discussion related to this topic was brief [14,15,34,35,36,39,42,43]. This can be considered as for the general recommendation in guidelines that, due to regional variations, a justification into the benefits, barriers, strengths or weaknesses was not appropriate as location may purely be based on limited options, finances or resources.

The topic of dosage in exercise interventions is usually related to the methods used to achieve the overload principle as part of a periodised training programme [49]. This will frequently relate to the FITT principles consisting of Frequency, Intensity, Type and Time [50]. Most guidelines discussed in some depth the use of dosage and methods of altering the dosage. This gives readers some generalisable guidance on how to progress exercise intervention in PwS. Interestingly, four of the guidelines [14,15,42,43] discussed the FITT principles for modifying exercise in PwS in more depth and gave strategies for making exercise more individualised. It was unclear in the remaining guidelines if there was a methodology based on the FITT principle for clinicians to follow, if modifying one component such as time or type of exercise resulted in an impact on the frequency, or if one component was more of a priority to progress over another. More interestingly, it was unclear in a majority of the guidelines how components such as intensity were assessed or measured as part of an initial fitness assessment. This in turn challenges the appropriateness of intensity prescription within exercise sessions, and the monitoring and evaluation at a later date. The lack of guidance on assessment methods resulted in poor guidance about appropriate outcome measures in exercise intervention. This fails to allow readers to select appropriate methods to quantify improvements to fitness as a result of exercise, or at which level of the WHO ICF framework these outcomes should be aimed [51]. 

Assessing the fidelity of an intervention provides the readers with a guide to the extent to which the intervention was delivered as planned and how to arrive at valid conclusions concerning its effectiveness [52,53]. Fidelity shares a strong relationship with long-term adherence in exercise intervention and, despite being acknowledged in a few of the guidelines [15,34,36], they all lacked considerable depth into how treatment fidelity is evaluated or monitored. This in turn suggests that exercise and activity groups for PwS are poorly or inconsistently attended, have a high dropout rate and will not provide an attractive platform for individuals to attend in the future. 

Possibly, due to the low occurrence of adverse events cardiac rehabilitation [54] and stroke [55], only four of the guidelines [15,35,42,43] demonstrated the use of pre-exercise safety checks. This covered a full medical review, electrocardiogram, heat rate, blood pressure and pulse oximetry. Although only one of these identified additional, during and post checks [15], all advocated the use of heart rate and blood pressure measurements due to their quick and effective use in clinical practice prior to commencing in exercise. 

A major limitation of this review is that the guidelines predominantly focus on the ambulatory stroke population, whereas the non-ambulatory population is potentially one of the most vulnerable and financially demanding groups. There is little mention of strategies to support long-term adherence to exercise interventions, and little exploration of the assessment and guidance of an individual’s level of self-efficacy [50]. It should also be acknowledged that CPGs provide guidance management for conditions as a whole. Therefore, priorities will be made based on best practice evidence, acute vs. subacute management, and long-term management of conditions.

This review identified the lack of guidance surrounding detailed exercise delivery and prescription in PwS. Specifically, the guidelines reviewed here are aimed at the ambulatory population and fail to address the non-ambulatory population, for which exercise as part of further stroke prevention, health life management and physical improvement is key. Future work needs to consider this population alongside the tailoring to needs of individuals, their specific exercise needs, and the safety checks that are needed to check exercise is ensuring their safety at all times.

## 5. Conclusions

To the authors’ knowledge, this is the first systematic review to examine CPG regarding exercise and physical activity interventions in PwS. 

CPGs for exercise and physical activity in PwS share similarities with general national physical activity guidelines in nonclinical populations. Unfortunately, CPGs for PwS lack depth and technical information on the actual exercise delivery and FITT principles. This creates a challenge for the target group (i.e., healthcare providers/exercise providers).

The lack of consistency with information related to the application, development, dosage and methods to progress exercise interventions for PwS provides a major barrier for healthcare practitioners. 

The success of exercise on referral schemes in the community is paramount. Schemes are under constant scrutiny for the effectiveness of their implementation and the intervention in clinical groups. There is a need to evaluate the success of exercise interventions, but we firstly need to consider how accessible the guidance is from CPGs. 

The take-home messages of this study are:Collectively well-known CPGs from the Global North are rigorous in their development, but only provide basic guidance on exercise in PwS.Specific information on exercise interventions is limited and requires clinicians to use their own experiences or knowledge, or to be aware of where to look for further guidance.The role of exercise in health is overwhelmingly strong, but proper implementation, reflection and evaluation of these services is imperative for long-term success and commissioning.

## Figures and Tables

**Figure 1 ijerph-19-01707-f001:**
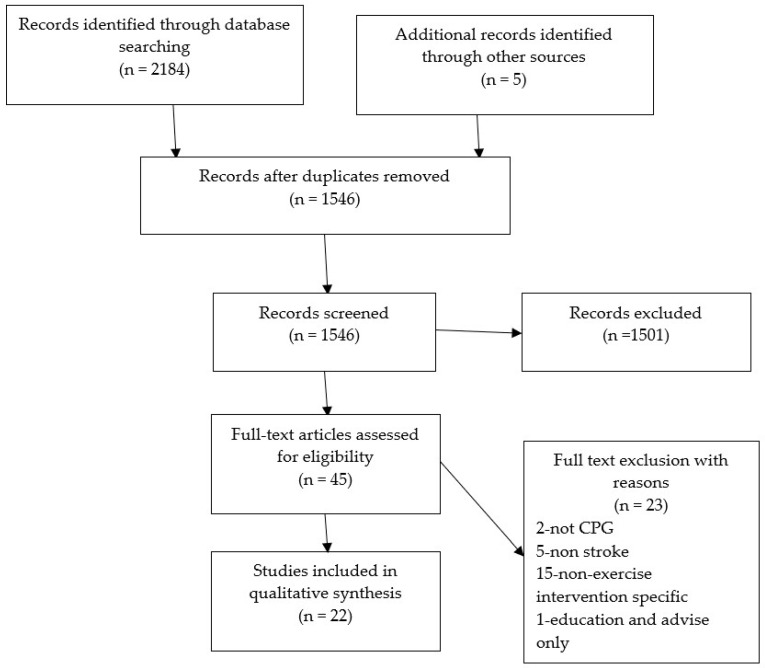
PRISMA flow diagram.

**Table 1 ijerph-19-01707-t001:** Selection criteria for search.

Exclusion Code	Category	Inclusion
1	Paper type	Clinical Practice Guidance including national and local/institution guidelines
2	Age of participants	18 years or older
3	Clinical group	Stroke
4	Intervention type	Exercise and exercise and rehabilitation
5	Intervention	Intervention
6	Language	English
7		No clear evidence from summary or abstract

**Table 2 ijerph-19-01707-t002:** Journal background information and types of guideline, grouped according to healthy lifestyle, prevention or rehabilitation, and year of publication.

Ref no	Guideline Name	Date	Publisher
	**Guidelines for Health**		
[31]	The National Physical Activity Plan: A Call to Action from the American Heart Association	2015	American Heart Association
[28]	Management of Stroke Rehabilitation	2019	VA/DoD Clinical Practice Guidelines
	**Guidelines for Prevention**		
[38]	Guidelines for management of ischaemic stroke and transient ischaemic attack 2008	2008	European Stroke Organisation
[27]	Guidelines for the Prevention of Stroke in Patients with Stroke and Transient Ischemic Attack (Secondary Stroke Prevention)	2014	American Heart Association
[37]	Canadian stroke best practice recommendations: secondary prevention of stroke, sixth edition practice guidelines	2018	CPG Infobase
	**Guidelines for Health and Prevention**		
[14]	Physical activity and exercise recommendations for stroke survivors: an American Heart Association scientific statement from the Council on Clinical Cardiology	2004	American Heart Association
[30]	Stroke Assessment Across the Continuum of Care	2005	Registered Nurses’ Association of Ontario
[29]	Primary prevention of ischemic stroke: a guideline from the American Heart Association/American Stroke Association Stroke Council	2006	American Heart Association
[39]	CPG for the Management of Stroke Patients in Primary Health Care	2009	GuiaSalud-Spanish guidelines
[41]	Clinical Practice Guidelines—Management of Ischemic Stroke 2nd edition.	2012	Stroke Council of the Malaysian Society of Neurosciences
[35]	Aerobic exercise after stroke	2013	Canadian Partnership for Stroke Recovery
[34]	Physical activity: exercise referral schemes: guidance (PH54)	2014	National Institute for Health and Care Excellence—NICE
[32]	Stroke and TIA	2017	NICE Clinical Knowledge Summaries
[40]	Stroke delivery plan: 2017 to 2020	2017	Welsh Government
[33]	Stroke and transient ischaemic attack in over 16s: diagnosis and initial management	2019	National Institute for Health and Clinical Excellence—Clinical Guidelines
	**Guidelines for Health, Prevention and Rehabilitation**		
[15]	Best Practice Guidance for the Development of Exercise after Stroke Service in Community Settings	2010	University of Edinburgh
[11]	Stroke rehabilitation in adults: guidance (CG162)	2013	National Institute for Health and Care Excellence—NICE
[42]	KNGF Clinical Practice Guideline for Physical Therapy in patients with stroke	2014	Royal Dutch Society for Physical Therapy (Koninklijk Nederlands Genootschap voor Fysiotherapie, KNGF)
[36]	Post Stroke Community Based Exercise Guidelines: A resource for community based exercise providers.	2015	Post Stroke Community Based Exercise Guidelines Working Group of the Ontario Stroke Network
[26]	AHA/ASA Guidelines for Adult Stroke Rehabilitation and Recovery	2016	American Heart Association
[10]	National clinical guideline for stroke: 5th edition	2016	Royal College of Physicians (RCP)
[43]	Clinical Guidelines for Stroke Management (Living Stroke Guidelines)	2020	Australian Stroke Foundation

**Table 3 ijerph-19-01707-t003:** Quality assessment using AGREE II tool.

Guideline	Publisher	Scope and Purpose				Stake Holder Involvement				Rigour of Development				Clarity of Presentation				Applicability				Editorial Independence		
		1	2	3	4	5	6	7	8	9	10	11	12	13	14	15	16	17	18	19	20	21	22	23
Aerobic exercise after stroke	Canadian Partnership for Stroke Recovery	6	5	5	5	3	6	5	5	5	5	5	5	7	1	5	5	7	5	5	1	1	1	1
AHA/ASA Guidelines for Adult Stroke Rehabilitation and Recovery	American Heart Association	7	7	6	6	6	7	3	5	5	6	3	6	5	1	6	6	7	1	2	1	2	6	6
Best Practice Guidance for the Development of Exercise after Stroke Services in Community Settings	University of Edinburgh	7	7	7	6	1	7	4	5	4	5	7	6	1	6	5	5	7	5	7	1	5	6	6
Canadian stroke best practice recommendations: secondary prevention of stroke, sixth edition practice guidelines		7	7	7	7	7	6	7	7	7	7	6	7	7	6	7	7	7	2	1	1	2	7	7
Clinical Guidelines for Stroke Management (Living Guidelines)	Australian Stroke Foundation	7	7	7	6	1	7	7	7	7	7	6	7	7	7	6	7	7	7	5	5	6	7	7
Clinical Practice Guidelines, Management of Ischemic Stroke, 2nd edition 2012.	Malaysian society of Neurosciences, Academy of medicine Malaysia, Ministry of Health malaysia)	6	6	7	2	1	7	1	5	7	6	5	5	7	7	6	6	6	6	2	2	6	1	1
CPG for the Management of Stroke Patients in Primary Health Care	GuiaSalud-Spanish guidelines	7	7	6	6	6	6	6	6	6	6	5	6	6	5	6	6	7	1	6	0	0	6	6
Guidelines for management of ischemic stroke and transient ischemic attack 2008	European Stroke Organisation	6	6	5	5	1	1	1	1	1	4	5	5	1	1	6	6	6	1	1	1	1	1	1
Guidelines for the Prevention of Stroke in Patients with Stroke and Transient Ischemic Attack (Secondary Stroke Prevention)	American Heart Association	7	7	6	6	1	7	5	6	5	6	5	6	6	1	6	6	7	6	2	6	2	6	6
KNGF Clinical Practice Guideline for Physical Therapy in patients with stroke	Royal Dutch Society for Physical Therapy	7	7	7	7	6	7	7	7	7	7	7	7	7	7	7	7	7	7	7	7	7	6	6
Management of Stroke Rehabilitation—VA/DoD Clinical Practice Guidelines	US Department of Veterans Affairs	6	6	6	6	6	6	5	6	6	6	6	7	6	6	6	6	7	6	5	5	1	7	7
National clinical guideline for stroke: 5th edition	Royal College of Physicians	6	6	6	6	6	6	5	6	6	6	6	7	6	6	6	6	7	6	5	5	1	7	7
Physical activity and exercise recommendations for stroke survivors: an American Heart Association scientific statement from the Council on Clinical Cardiology	American Heart Association	6	6	7	3	1	1	1	1	1	1	5	5	1	1	6	1	7	1	1	1	1	1	1
Physical activity: exercise referral schemes: guidance (PH54)	National Institute for Health and Care Excellence	7	7	6	7	7	7	7	7	7	7	6	6	7	6	6	5	7	7	7	7	6	7	7
Post Stroke Community Based Exercise Guidelines A Resource for Community Based Exercise Providers	Ontario Stroke Network	7	6	6	6	6	7	5	2	1	1	5	6	7	5	6	6	6	2	5	1	2	6	6
Primary prevention of ischemic stroke: a guideline from the American Heart Association/American Stroke Association Stroke Council	American Heart Association	7	7	7	7	1	6	6	6	6	6	7	7	7	1	7	7	7	4	1	1	6	7	7
Stroke and TIA Clinical Knowledge Summaries	National insitiute for Health and Clinical Excellence	7	7	7	7	7	7	7	7	7	7	6	7	7	7	7	7	7	1	3	3	7	7	7
Stroke and transient ischemic attack in over 16s: diagnosis and initial management	National Institute for Health and Clinical Excellence	7	7	7	7	7	7	7	7	7	7	6	7	7	7	7	7	7	1	6	4	7	7	7
Stroke Assessment Across the Continuum of Care	Registered Nurses’ Association of Ontario	7	7	7	5	6	7	3	1	5	1	1	6	6	7	7	6	6	6	6	1	6	5	5
Stroke delivery plan 2017 to 2020	Welsh Government	7	7	7	7	1	7	1	1	1	1	5	1	7	6	7	6	5	1	6	4	5	4	4
Stroke rehabilitation in adults: guidance (CG162)	National Institute for Health and Care Excellence	7	7	7	7	7	7	7	7	7	7	6	7	7	7	7	7	7	1	6	4	7	7	7
The National Physical Activity Plan: A Call to Action from the American Heart Association	American Heart Association	6	6	5	7	1	6	1	1	1	5	5	6	7	1	5	5	6	1	1	1	1	7	7

**Table 4 ijerph-19-01707-t004:** Data extraction using CERT and safety data.

Guideline	Publisher	Domains from CERT																					
		What: Materials	Who: Provider											Where: Location	When, How Much: Dosage	Tailoring: What, How			How Well, Planned, Actual		Pre Checks	During Checks	Post Checks
		1	2	3	4	5	6	7a	7b	8	9	10	11	12	13	14a	14b	15	16a	16b			
Aerobic exercise after stroke	Canadian Partnership for Stroke Recovery	1	0.5	0.5	0.5	0.5	0.5	0.5	0.5	0	0.5	0.5	0	0.5	0.5	0.5	0.5	0.5	0	0	0.5	0	0
AHA/ASA Guidelines for Adult Stroke Rehabilitation and Recovery	American Heart Association	0.5	0	0.5	0	0	0	0	0	0	0	0	0	0	0.5	0.5	0.5	0	0	0	0	0	0
Best Practice Guidance for the Development of Exercise after Stroke Service in Community Settings	University of Edinburgh	1	1	1	0.5	0.5	0.5	0.5	0.5	0.5	0.5	0.5	1	0.5	1	1	1	1	0.5	0	0.5	0.5	0.5
Canadian stroke best practice recommendations: secondary prevention of stroke, sixth edition practice guidelines	CPG Infobase	0.5	0.5	0	0	0	0	0	0	0	0	0	0	0	0.5	0	0	0	0	0	0	0	0
Clinical Guidelines for Stroke Management (Living Guidelines)	Australian Stroke Foundation	0.5	0	0.5	0.5	0	0	0	1	0.5	0.5	0.5	0.5	1	0.5	0.5	0.5	0	0	0.5	1	0	1
Clinical Practice Guidelines, Management of Ischemic Stroke, 2nd edition 2012.	Malaysian society of Neurosciences, Academy of medicine Malaysia, Ministry of Health malaysia)	0	0	0	0	0	0	0	0	0	0	0	0	0	0	0	0	0	0	0	0	0	0
CPG for the Management of Stroke Patients in Primary Health Care	GuiaSalud- Spanish guidelines	0.5	0	0.5	0	0	0	0	0	0	0	0	0	0.5	0	0.5	0	0	0	0	0	0	0
Guidelines for management of ischemic stroke and transient ischemic attack 2008	European Stroke Organisation	0	0	0	0	0	0	0	0.5	0	0	0	0	0	0	0.5	0.5	0	0	0	0	0	0
Guidelines for the Prevention of Stroke in Patients with Stroke and Transient Ischemic Attack (Secondary Stroke Prevention)	American Heart Association	0.5	0.5	0	0	0	0	0	0	0	0	0	0	0	0.5	0.5	0	0	0	0	0	0	0
KNGF Clinical Practice Guideline for Physical Therapy in patients with stroke	Royal Dutch Society for Physical Therapy	1	0.5	0.5	1	0.5	0	1	1	0	0.5	1	0.5	1	0.5	1	1	0.5	0.5	0.5	1	0	1
Management of Stroke Rehabilitation- VA/DoD Clinical Practice Guidelines	U.S Department of Vetrans Affairs	0.5	0	0.5	0	0	0	0	0	0	0	0	0	0	0.5	0	0	0	0	0	0	0	0
National clinical guideline for stroke: 5th edition	Royal College of Physicians	0.5	0.5	0.5	0	0	0	0	0	0	0	0.5	0	0	0	0.5	0.5	0.5	0	0	0	0	0
Physical activity and exercise recommendations for stroke survivors: an American Heart Association scientific statement from the Council on Clinical Cardiology	American Heart Association	1	0	0.5	0.5	1	0.5	1	1	0	0	0	0.5	0.5	1	1	1	0.5	0	0	0	0	0
Physical activity: exercise referral schemes: guidance (PH54)	National Institute for Health and Care Excellence	1	0	0.5	0	0.5	0.5	0	0.5	0	0.5	0.5	0	0.5	0.5	0.5	0	0	0.5	0	0	0	0
Post Stroke Community BasedExercise Guidelines. A Resource for Community Based Exercise Providers	Ontario Stroke Network	1	1	0.5	0.5	0.5	0	0.5	0.5	0.5	0.5	0.5	0.5	0.5	0.5	1	1	0.5	0.5	0	0	0	0
Primary prevention of ischemic stroke: a guideline from the American Heart Association/American Stroke Association Stroke Council	American Heart Association	0.5	0	0	0	0	0	0	0	0	0	0.5	0	0	0.5	0.5	0	0	0	0	0	0	0
Stroke and TIA—Clinical Knowledge Summaries	National Institute for Health and Clinical Excellence	0.5	0	0	0	0	0	0	0	0	0	0.5	0	0	0	0.5	0	0	0	0	0	0	0
Stroke and transient ischemic attack in over 16s: diagnosis and initial management—Clinical Guidelines	National Institute for Health and Clinical Excellence	0	0	0	0.5	0	0	0	0	0	0	0	0	0	0.5	0.5	0.5	0	0	0	0	0	0
Stroke Assessment Across the Continuum of Care	Registered Nurses’ Association of Ontario	0.5	0	0	0	0	0	0	0	0	0	0	0	0	0.5	0	0	0	0	0	0	0	0
Stroke delivery plan 2017 to 2020	Welsh Government	0.5	0	0	0	0	0	0	0	0	0	0	0	0	0.5	0	0	0	0	0	0	0	0
Stroke rehabilitation in adults: guidance (CG162)	National Institute for Health and Care Excellence	1	0.5	0	0	0	0	0.5	0.5	0	0	0	0	0	0.5	0.5	0.5	0	0	0	0	0	0
The National Physical Activity Plan: A Call to Action from the American Heart Association	American Heart Association	0.5	0	0	0	0	0	0	0	0	0	0	0	0	0.5	0.5	0	0	0	0	0	0	0

## Data Availability

Not applicable.

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
