# Peer review of "Examining Clinical Practice Guidelines for Exercise and Physical Activity as Part of Rehabilitation for People with Stroke: A Systematic Review"

_ijerph, 2022, doi:10.3390/ijerph19031707_

Round 1

Reviewer 1 Report

General Comments

General Weaknesses

- Manuscript must present the practical guidelines to prescribe exercise for people with Stroke (a table must be elaborated with practical and useful data for proffesionals and researchers).

- Despite Introduction is interesting, it must be improved. The rationale of a systematic review to assess the manuscript´s topic must be reforced (three or four ideas to justify why is important to carry out a systematic review about this topic). Addiotinally, the manuscript´s aims must be presented in a paragraph (not in a set of  “points”) (lines 141-159).

- Discussion must be improved. Its structure must be elaborated following the interanational standards for this type of manuscript (I recommend to consult the PRISMA guidelines and some papers to do this. For instance, the first paragraph must present the main study´s goal and the main study´s results. Subsequently, the whole results of the manuscript must be discussed (these results must be compared with those of the previous studies. In addition, “the underlying mechanisms” of the study´s results must be also presented. Finally, the study´s limitations must be located in the last paragraph of the Discussion (it must be included in Discussion) (lines 484-606).

- This section must be elaborated again (Conclusions). It must be linked to the study´s goals and it must be brief (4-5 lines)

General Strengths

- This section is (Results) very interesting. It contents abundant useful data for readers and researchers (pages 6 of 12).

Major Comments:

Title

Strengths

- Title is pertinent and correct (lines 1-4).

Abstract

Weaknesses

- Abstract must be elaborated again. It must follow recommendations of the present report. It must be brief (please, consult the journal´s guidelines) (lines 20-61).

Keywords

Weaknesses

- Keywords must be corrected (please, avoid using the same words in the title and in the keywords) (lines 62).

Introduction

Weaknesses

- These kind of sub-sections must not be used in the Introduction (1.1. Guidelines of exercise and physical… (line 80),1.2. Guidelines of exercise in clinical… (line 100).  

- Despite Introduction is interesting, it must be improved. The rationale of a systematic review to assess the manuscript´s topic must be reforced (three or four ideas to justify why is important to carry out a systematic review about this topic). Addiotinally, the manuscript´s aims must be presented in a paragraph (not in a set of  “points”) (lines 141-159).

Methods

Weaknesses

- Despite of Methods is correct, it demands improvements. Search strategy and 2.2. Databases sub-sections and Analysis and synthesis must be corrected; they must be presented in a regular paragraph (please, don´t divide the information in parts; e,g.,a), b) and c) or something similar). Additionally, Tables 2 must be deleted (lines 194-223 and 305-314, respectively).

Results

Strengths

- This section is (Results) very interesting. It contents abundant useful data for readers and researchers (pages 6 of 12).

Weaknesses

- The Prisma flow diagram must be corrected. The box entitled “Full-text articles excluded, with reasons… must be reduced (at the botton) (lines 334-371).

- Table 3 must be corrected. It must be elaborated with the same font the text (lines 394-398). Please, correct the whole similar format mistakes. 

Discussion

Weaknesses

- Discussion must be improved. Its structure must be elaborated following the interanational standards for this type of manuscript (I recommend to consult the PRISMA guidelines and some papers to do this. For instance, the first paragraph must present the main study´s goal and the main study´s results. Subsequently, the whole results of the manuscript must be discussed (these results must be compared with those of the previous studies. In addition, “the underlying mechanisms” of the study´s results must be also presented. Finally, the study´s limitations must be located in the last paragraph of the Discussion (it must be included in Discussion) (lines 484-606).

Conclusions

Weaknesses

- This section must be elaborated again. It must be linked to the study´s goals and it must be brief (4-5 lines)

References

- This section must be checked it in details. It could contain format mistakes (lines 674-809).    

Tables and Figures

- Please, see previous comments about Tables and Figures.

Author Response

Word document attached with table of responses 

Reviewer 2 Report

Thank you for allowing me to read this paper which initially deals with an interesting and relevant topic. The suggestions given in this document are intended to improve your work. The same feedback document will be given to both editors and authors.

1) I hope the authors will forgive my frankness, but it is a pity that such an interesting work for several rehabilitation groups is so poorly presented. Please re-read the journal's guidelines carefully https://www.mdpi.com/journal/ijerph/instructions. For example:

  • Information on authors' affiliations is not well presented.
  • Authors should complete the back matter.
  • The abstract should be no longer than 200 words and should not use the headings described above.
  • Tables and figures do not follow the journal guidelines and must be revised and redone.
  • There are blank pages, spacing and text that do not correspond to the IJERPH template.
  • Check the list of references to ensure that it matches the requirements of the journal, adding the DOI of the publications that have such an identification.

2) Although the authors claim to have followed the PRISMA statement, I have my doubts. The authors refer to the PRISMA statement from 2009, when there is an update from 2020. And the PRISMA diagram is not updated either. Please upload THE PROPERLY COMPLETED PRISMA CHECKLIST AS SUPPLEMENTARY MATERIAL to check that you comply with the declaration, and an explanation for the items that do not comply. You can find all the information here http://prisma-statement.org

Additional comments:

  • The introduction should define all the variables to be taken into account in the work. It can be improved.
  • The objective of the review could be better written, and a PICO question is usually used in systematic reviews. It can be better written.
  • Exclusion criteria cannot be the negation of inclusion criteria.
  • The methods section needs to be restructured (the recommended readings will help).
  • Table 3 needs to be much more complete.
  • Where is the risk of bias analysis?
  • There is mixed reporting of methods and results.
  • The discussion needs to be almost completely redone. Present your main results, then contrast some studies with others and draw out the study variables you consider in your objective (which is not clear to me), state the practical implications of your study, and end with limitations and future directions.
  • Check that the conclusions are really conclusions.

 In sum, I recommend that authors READ the journal guidelines and the PRISMA statement thoroughly, CHEK both formatting and writing, and REWRITE their manuscript so that I can properly review it.

Author Response

(The authors gave the same response as above.)

Round 2

Reviewer 1 Report

General Comments

General Weaknesses

Discussion

Weaknesses

- These kind of sub-sections must not be used in the Discussion or any manuscript section (4.1. [line 626], 4.2. [line 638], etc.). Please, correct this at the whole manuscript.   

Conclusions

Weaknesses

- This section must be numbered like this; 5. Conclusion (line 821).      

Author Response

Rebuttal table attached 

Reviewer 2 Report

Considering the above, I can accept the paper as the authors have greatly improved its quality, even though the exclusion criteria are still the negation of the inclusion criteria and there is no analysis of the risk of bias. The rest seems to be in order.

Author Response

Rebuttal table attached